# Atmospheric Photoionization Detector with Improved Photon Efficiency: A Proof of Concept for Application of a Nanolayer Thin-Film Electrode

**DOI:** 10.3390/s21227738

**Published:** 2021-11-20

**Authors:** Adelaide Miranda, Pieter A. A. De Beule

**Affiliations:** International Iberian Nanotechnology Laboratory, Avenida Mestre José Veiga s/n, 4715-330 Braga, Portugal; adelaide.miranda@inl.int

**Keywords:** photoionization, photoionization detector, volatile organic compounds, gas sensing, gas sensor, nanolayer thin film

## Abstract

Atmospheric photoionization is a widely applied soft ionization mechanism in gas sensing devices for the detection of volatile organic compounds in ambient air. Photoionization is typically induced by low-pressure Vacuum Ultraviolet (VUV) lamps with MgF_2_ or LiF lamp surface windows depending on the gas fill and the required wavelength transmission window. These lamps are known to exhibit gradually reduced VUV transmission due to hydrocarbon contamination. LiF surface windows are known to be especially problematic due to their hygroscopic nature, reducing VUV lamp lifetime to a mere 100 h, approximately. Here, we present a new design for the electrode of a photoionization detector based on thin-film technology. By replacing the commonplace metal grid electrode’s VUV lamp surface window with a chromium/gold thin film we obtain a doubling of photon efficiency for photoionization. Replacing the hygroscopic LiF lamp window surface with a metallic layer additionally offers the possibility to vastly increase operational lifetime of low-pressure Argon VUV lamps.

## 1. Introduction

Atmospheric Pressure Photoionization (APPI) is a well-known soft ionization mechanism that induces the formation of electron-ion pairs from molecules upon the absorption of high energy photons, usually without disintegrating the molecules into multiple fragments. APPI is applied in several trace gas or liquid analysis detection systems, including photoionization detectors (PID) [1], mass spectrometry [2], ion mobility spectrometry [3,4,5] and chromatography [6,7,8]. The lowest binding energy of an electron to a molecule, known as the first Ionization Potential (IP), typically lies within the range of 5 to 25 eV. Hence, high energy photons are required to ionize molecules. These photons are typically produced by a low-pressure gas discharge lamp that emits intense light between 105 nm (11.8 eV) and 150 nm (8.4 eV) depending on the low-pressure gas fill. Most frequently encountered gas fills are Xenon (Xe, 8.4 and 9.6 eV), Deuterium (D, 10.2 eV), Krypton (Kr, 10.6 eV) or Argon (Ar, 11.8 eV). At such photon energies light is commonly referred to as vacuum ultraviolet (VUV) because it is absorbed by air with an oxygen component (. The IPs of water (12.6 eV), acetonitrile (12.2 eV), nitrogen (14.5 eV) and helium (23 eV) are all considerably greater than the photon energies of VUV lamps, rendering them insensitive to light, while most organic molecules exhibit IPs between 7 and 10.5 eV. This makes VUV lamps ideal ionization tools for the detection of a wide variety of molecular compounds through the fact that the IPs of most carrier gasses and carrier liquids lie above the photon energy of VUV lamps.

VUV lamps can be divided in two categories depending on the electric field used to excite the low-pressure gasses: lamps excited with electric fields composed of discharge (DC), or radio (RF) frequencies. Lamps operated with DC fields consist of a metallic cathode and anode within a glass enclosure, between which a high voltage of around 1 kV is applied to start the VUV generating discharge. These lamps are used in gas chromatography and tend to create a narrow and focused output beam. Lamps driven by RF frequencies, on the other hand, simply consist of a transparent glass enclosure containing the low-pressure gas and can be further divided subcategories depending on the coupling of RF power to the low-pressure gas container of the VUV lamp, i.e., capacitively or inductively. RF VUV lamps can be made smaller, and exhibit a higher energy efficiency, than DC VUV lamps. They are therefore the preferred VUV photon source for portable gas detectors. RF excited lamps tend to emit an unfocussed collimated light beam. Common silicon-based glasses do not transmit VUV. Hence, a crucial component of the VUV lamps described above is the VUV transmission window. Magnesium fluoride (MgF_2_) and lithium fluoride (LiF) crystals are widely used materials for these transmission windows. MgF_2_ is a frequently encountered material in infrared (IR) optics, but also transmits UV light down to 110 nm. It is also known as a rugged material resistant to chemical etching, laser damage, and mechanical and thermal shock. This makes it the transmission window material of choice for most VUV lamps. One notable exception is the high energy VUV lamp based on argon that requires LiF crystals because a sufficiently high transmission coefficient for light below 110 nm is required. However, LiF has several disadvantages. First, LiF is a highly hygroscopic material, i.e., it dissolves in highly humid environments. This is a major contributing factor to short operational lifetimes, sometimes down to 100 h. A second major disadvantage of LiF is the incompatibility of its thermal expansion coefficient with ordinary glass, making it very hard to obtain the good seal that is necessary to avoid contamination or water in the gas enclosure of the lamp. [9]

Continuous APPI effected with the VUV lamps described above results in a gradual reduction of the transmission efficiency of the VUV lamp windows due to adhesion of hydrocarbons [10,11]. Nanometer sized thin films of hydrocarbons are sufficient to cause a significant drop in VUV intensity. This phenomenon can be partly mediated by cleaning the lamp window with methanol. However, operation of the VUV lamp in high concentrations of volatile organic compounds can create such an intense deposition of hydrocarbons on the lamp window that cleaning with methanol does not suffice, and a treatment with high purity alumina (Al_2_O_3_) particles or fine diamond polish becomes necessary [12]. Such cleaning procedures eventually lead to the gradual degradation and eventual destruction of the lamp window.

Here, we investigate the potential of thin-film technology to replace metallic grids integrated within PIDs. We propose and apply a new type of electrode, consisting of a gold and chromium nanolayer, for generation of the electric field required for capturing the ionized gas vapors effectively. We compare the performance of an experimental rig with the current state-of-the-art using different toluene vapor concentrations. We conclude by discussing the proposed technology.

## 2. Materials and Methods

### 2.1. Materials, Instruments and Software

For the set-up of the PID electrode we used a high-voltage power supply PS350 from Stanford Research Systems, Sunnyvale, CA, USA; a transimpedance amplifier IVC102 from Burr-Brown, Tucson, AZ, USA; a conductive O-ring from Parker Chomerics, Woburn, MA, USA; a commercial PID sensor piD-TECH from AMETEK MOCON, Brooklyn Park, MN, USA; a bare MgF_2_ glass window purchased from AOTK, Xiamen, China; a Krypton gas fill from PKR 106-6, Heraeus Noblelight, Banbury, Great Britain; a National Instruments CompactDAQ data acquisition platform cDAQ-9178 chassis with modules NI 9401 and NI 9201, from National Instruments, Austin, TX, USA, and two independent gas lines from GFC mass flow controller, Aalborg Instruments & Controls, Orangeburg, New York, NY, USA. For the acquisition of data we used LabVIEW LabVIEW 2011, National Instruments, Austin, TX, USA, with a software routine developed by the authors and available at the GitHub (https://github.com/INLnano/LabVIEW).

### 2.2. Set-Up

Figure 1 shows a representation of our in-house PID set-up and implementation. An ionization volume was created within a stainless-steel enclosure, optimal for contamination removal through a bake-out procedure in an oven avoiding memory effects in the measurements reported. An electric field is generated between two opposing electrodes (electrode 1 and 2) through a 50 V DC signal applied to electrode 1 by a high-voltage power supply. The opposing electrode 2 collects the low current PID signal that is subsequently amplified with an integrator transimpedance amplifier configured in a basic reset and integrate set-up. This amplifier integrates low-level input currents a user-determined period, storing the resulting voltage on an integrating capacitor. The output voltage is held for accurate measurements and provides a lower-noise alternative to conventional op-amp circuits encountered in commercial PID technology.

Figure 2a,b shows a close-up of the PID ionization chamber and sensor electrodes applied in this investigation. The ionization chamber volume is bound by O-rings onto which the detection electrode with corresponding electronic amplification circuit is pressed with screws, while a conductive O-ring is used for application of an ionization current collection voltage on electrode 2. Figure 2c shows a typical PID sensor electrode from a commercial PID sensor consisting of a metallic plate with 19 holes with a 525 μm diameter, corresponding to a total surface area of ~4.1 mm^2^. Taking into consideration that the VUV lamp surface covering this electrode has a surface area of 15.9 mm^2^ and that the RF lamp emission is unfocused, only ~26% of the VUV lamp emission is used. Figure 2d shows a photograph of the thin film stack on a bare MgF_2_ glass window consisting of a 2.5 nm of chromium (Cr) and a 2.5 nm layer of gold (Au) adhesion layer that was deposited in a magnetron sputtering in an Alcatel SCM-450, as described in [13]. This design resulted in an ionization chamber volume of ~425 mm^3^. 

### 2.3. Toluene Gas Vapour Measurements

Toluene vapour concentrations of 0.7, 1.4, 2.1 and 2.8 ppm within a constant nitrogen flow of 200 mL/min were exposed to the in-house PID set-up described above. The VUV lamp has a low-pressure Krypton gas fill which emits light with photon energy of 10.6 eV, which is substantially above the toluene ionization potential of 8.828 ± 0.001 eV [14].

Digital control signals S1 and S2, determining reset, start and stop of the integration period, as well as the V0 voltage readout of the transimpedance amplifier, were generated (digital output) and read (analog input) using a National Instruments CompactDAQ data acquisition platform controlled through LabVIEW. The PID voltage output values were obtained after integrating the transimpedance amplifier at 200 ms.

Two stainless steel thermal mass flow controllers (MFCs) were used to indicate and control nitrogen gas flow rates in two independent gas lines. One of the gas lines was passed through a bottle of liquid toluene to acquire a saturated vapour pressure concentration (29.1 ppm at 20 °C laboratory temperature). This line was then merged with the other nitrogen gas line by a T-junction. Varying the gas flow rates of the two gas lines, we could obtain software-controllable concentrations of toluene in a constant background of 200 mL/min nitrogen. As with the transimpedance amplifier, the MFCs were controlled through the National Instruments CompactDAQ data acquisition platform controlled through LabVIEW. A software routine in LabVIEW was written to generate five consecutive exposures of 200 s each of 0.7, 1.4, 2.1 and 2.8 ppm toluene concentrations that are well below the 8 hour average Permissible Exposure Level (PEL) of 200 ppm as defined by the Occupational Safety & Health Administration (OSHA) of the United States Department of Labor [15].

## 3. Results and Discussion

We compared the performance of a standard PID electrode with a nanolayer thin-film electrode in terms of ionization efficiency. We achieved this by compressing a low-pressure Kr VUV lamp with MgF_2_ transmission window against a bare MgF_2_ substrate and MgF_2_ coated with Cr (2.5 nm)/Au (2.5 nm) thin film, respectively. This avoided the requirement to deposit the Cr/Au nanolayer directly on the VUV window. It would however be desirable for commercial applications, since it avoids the integration of a superfluous MgF_2_ interface. In future, this could, for example, be addressed by mounting a thin film-coated MgF_2_ substrate directly on the lamp during fabrication. In the absence of the Cr/Au nanolayer stack, an electric field for collection of the photoionization current was created by placing two small copper strips and applying the 50 V DC to it as with the Cr/Au nanolayer stack. Similar copper strips were applied to the Cr/Au coated window to enable comparison of the photoionization current between coated and non-coated MgF_2_ glass windows, providing an indirect measurement of VUV light transmission efficiency of the Cr/Au nanolayer stack.

Figure 3 details the response of the PID with a Krypton VUV lamp that has a MgF_2_ window with and without the Cr/Au nanolayer stack at five consecutive exposures of 200 s each of 0.7, 1.4, 2.1 and 2.8 ppm toluene. At low concentrations of <1 ppm, the PID system reported a stable voltage output, while higher concentrations resulted in a response that gradually increased over time. A 200 s, the PID sensor recovery time was integrated after each exposure with toluene. For higher concentrations, significant pulsing was observed. We obtained a constant response for 0.7 ppm toluene for both bare MgF_2_ windows and the same windows coated with the 2.5 nm Cr/Au thin film nanolayer. The Cr/Au-coated window exhibited a 52.6% photoionization current with respect to the bare MgF_2_ window. Doubling the concentration to 1.4 ppm increased the voltage output of the transimpedance amplifier from 1.05 V to 1.51 V, indicating a partial saturation effect within our in-house PID design. This saturation was also manifested by a slightly increasing amplifier readout over the timeframe of the exposure with 1.4 ppm of toluene. This was also reflected in an increase of the relative photoionization current between bare and Cr/Au-coated windows from 52.5% (0.7 ppm) to 53.8% (1.4 ppm), 63.4% (2.1 ppm) and 73.9% (2.8 ppm) respectively. This increase cannot be explained by an improved transmission of VUV through the coated layer as this was expected to be constant, but rather must be attributed to saturation effects.

## 4. Conclusions and Outlook

In this study, we proposed and applied a new type of electrode for generation of the electric field required for capturing the ionized gas vapours effectively. We deposited a thin film Cr (2.5 nm)/Au (2.5 nm) stack on a bare MgF_2_ glass window and compared the photoionization performance of a home-built PID with a bare MgF_2_ glass window, providing an indirect measurement of the VUV transmission efficiency of the thin film at the VUV wavelength produced by a low-pressure Krypton lamp. We observed >50% transmission efficiency, which is related to the 26% utilization of the VUV lamp emission in a state-of the art PID sensor. This implies that we achieved a doubling in photon efficiency within the PID sensor within this proof-of-concept experiment. It should be noted that for different thin-film materials, potentially thinner films, and different VUV wavelengths from Xe, D or Ar-based lamps, and different photon efficiency gains can be expected due to light dispersion characteristics [16]. Aside from the photon efficiency improvement, changing the chemistry of the lamp window surface of PIDs relying on hygroscopic LiF windows of Argon filled VUV lamps can be hugely beneficial for increasing operational lifetime.

Future research will focus onto the design of VUV lamp coatings that can strongly reduce hydrocarbon contamination linked to degradation of PID performance. Through material composition optimization of the nanolayer stack on the lamp windows, a balance will need to be struck between minimal impact on VUV lamp light transmission and maximal efficiency in avoiding hydrocarbon deposition during PID deployment. Especially in environments where it is impossible to clean or replace VUV lamps, a contamination resistant PID sensor is expected to be of great value for increased in-the-field use of PID technology.

## Figures and Tables

**Figure 1 sensors-21-07738-f001:**
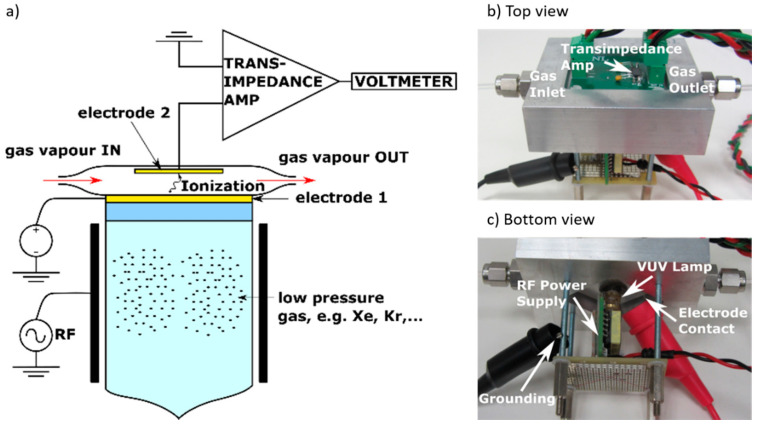
(**a**) PID sensor schematic lay-out; (**b**) PID sensor set-up aluminum block with gas inlet and outlet along with electronics circuit board comprising electrode 2 and transimpedance amplifier circuit; (**c**) VUV lamp comprised electrodes conveying excitation electric power to a miniature Krypton-filled VUV lamp.

**Figure 2 sensors-21-07738-f002:**
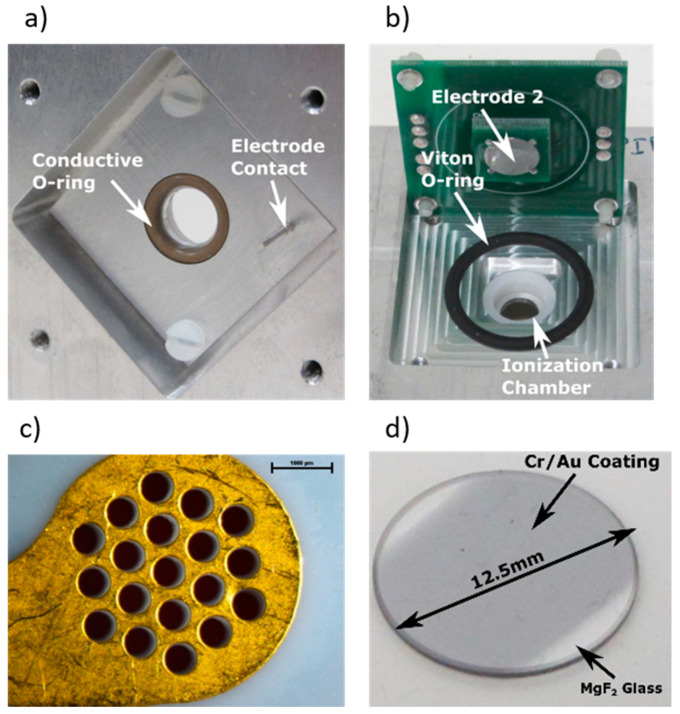
In-house fabricated PID ionization chamber view from (**a**) top and (**b**) bottom, and PID sensor electrode (**c**) state-of-the-art PID reference electrode, scale bar 1000 µm and (**d**) PID reference electrode fabricated through deposition of a Cr (2.5 nm)/Au (2.5 nm) sputtered thin films.

**Figure 3 sensors-21-07738-f003:**
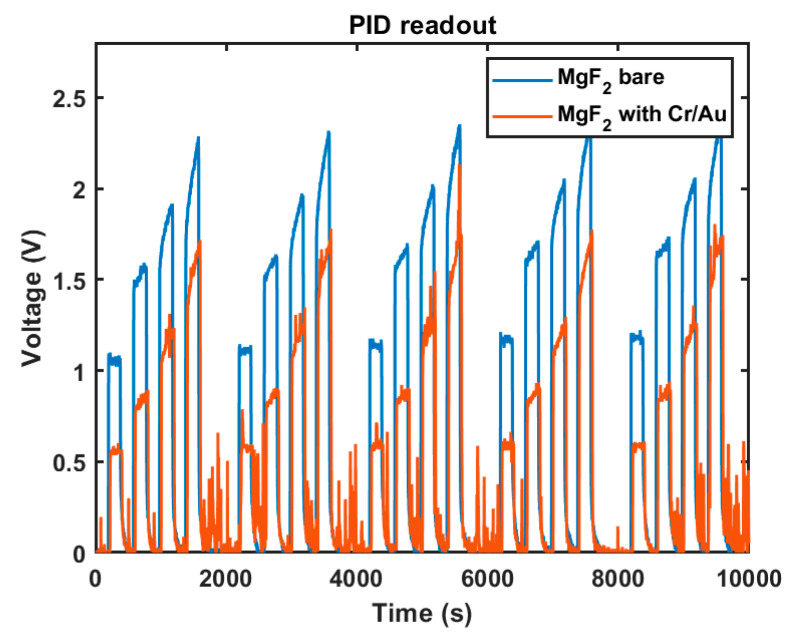
PID sensor readout with Cr/Au nanolayer stack (orange line) and using a bare MgF_2_ window (blue line).

## Data Availability

Not applicable.

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
