# Peer review of "Atmospheric Photoionization Detector with Improved Photon Efficiency: A Proof of Concept for Application of a Nanolayer Thin-Film Electrode"

_sensors, 2021, doi:10.3390/s21227738_

Round 1
Reviewer 1 Report
Although the experimental set-up is sound and the topic/implementation is indeed interesting, there are a lot of things that need to be improved for this paper to reach publication stage. While more details are highlighted in the annotated manuscript for the benefit of the authors, here is a rough outline of the main points that require attention:
- The text is rife with syntactical and grammatical errors, starting with the title itself. These have been 99% highlighted in the annotated pdf, which the authors are called to correct.
- The references are too few, too scarce, and too outdated (most recent one from 2014) - they are also not in the format espoused by the journal. The authors are advised to rewrite the introduction by trying to present at least a few examples of recent contributions to the topic, discussing either PPID, VUV lamps, or an overview of thin film applications in optics, for example). Already in the present text, there are multiple instances where referencing could be beneficial, even if that might be a textbook reference for fundamental issues.
- The basic premise of the paper, i.e. the thin film Cr/Au bilayer coating on the MgF2 window, has not been discussed at all. How has this been prepared? In the Acknowledgements, we read that this was prepared by a INESC professor - if there is an issue with co-authorship, then a simple outline and discussion of the method would suffice. However, such information is important. Even if the deposition method is straightforward, some referencing is also needed for the reader to recreate the experiment.
- Apart from the photos and schematics, the paper essentially contains only one result, presented in Figure 3, which also needs to be improved for clarity of presentation. What other results and metrics can be extracted by the experimental setup? For example, the authors are mentioning photoionization currents, transmission efficiencies etc. Where are these presented or calculated (and how)? Another proposal is to present the LAbView VI that was built for this experimental setup.
- Other topics that are presented in the abstract/introduction and conclusions that are not been discussed or shown in the results are: the hygroscopicity of materials, the limitations and prolonging of lifetime, hydrocarbon contamination etc. Some relevant content needs to be discussed and presented on these, even if it's simple photos (I don't expect the authors to divert into Materials Science, but still there are much that can be included to increase the value of this paper).

Reviewer 2 Report
The authors applied a layer of thin metal electrode to improve photo efficiency for atmospheric photoionization detector. The testing experiments showed that the thin layer with 2.5 nm Cr and 2.5 nm Au could double the photo efficiency as compared to the traditional one using metal strip. The purpose is quite interesting and it has wide application in gas chromatography. However, the underlying mechanism to improve the photo efficiency is not clear. Is it due to the increase of electrode surface? If that is the case, a metal strip with larger surface would also work. The experiment data are too less for a scientific paper, the influence of thin film thickness and material, the sensitivity, the response time, the linear range etc. are all missing. In addition, the photo efficiency of commercialized PID is higher than the one in this manuscript (the detection limit is much lower than 0.7ppm). Then, what kinds of electrodes they use and how to further improve their efficiency is more important. Therefore, it cannot be published in the current form and should be further improved.
1.MgF2 thin film has a low refractive index, which will improve the optical efficiency to some extent. The author sputtered a layer of gold film on the MgF2 glass surface. And it can be seen Cr/Au nanolayer stack photon efficiency is better than that bare MgF2 thin film in the Figure 3. Theoretically, what will be the photon efficiency changed-improve constantly or declined or invariable if a layer of other materials with low refractive index is coated on its surface? Meanwhile, what are the advantages of gold film than others?
- What is the mechanism of improving photo efficiency after sputtering Cr/Au film? Through the application of metal thin film,why the photo efficiency for photoionization can be greatly improved?
- In published research paper, the detection capability of sensors has reached ppb level. However, the sensor designed by the authors detection capability can only be achieved ppm level even after sputtered Au film. What is the core significance of designing this sensor?
- The performance of the new sensor constructed in this paper is not comprehensive. Although the Au film is sputtered on the MgF2 glass surface, the photo efficiency is improved, but does the other performance of the sensors will change? For example: sensitivity, response time, linearity, the operational lifetime of the device and so on.
- The authors just listed too few references. The underlying science and their contributions compared to the references are not clear. The authors should focus more on the latest progress of the work and cite rather than expressing the basics. Meanwhile, the gramma of the text should be checked carefully, e.g. Line 176, 177 on Page 5, “applications since at avoids the integration of an extra superfluous MgF2 interface”, Line 191 on Page 5, “a significant amount of after pulsing”.
- Where do the percentages of photoionization current increase come from - 52.5% (0.7 ppm) to 53.8% (1.4 ppm), 63.4% 199 (2.1 ppm) and 73.9% (2.8 ppm)? It is not clear how to calculate. Also how to obtain “26 % of VUV the lamp emission” data? Did the authors conduct any experiment?
- The inset annotations are not correct in Figure 3 (the blue line should be MgF2 with Cr/Au).
Round 2
Reviewer 1 Report
My apologies to the authors for not finding the annotations to the comments made on the pdf document. I clearly remember spending a considerable time on this review - I don not know why my comments were not saved. Nevertheless, it is appreciated that the authors took the time to go through the highlighted text in order to correct whatever could be corrected based on the (lack of) information provided.
The provided explanations and added references have covered my original remarks and helped to present a better integrated version of the original. I am recommending its Acceptance for Publication.
PS: compliments for the James Lovelock reference. The man is a legend.
Reviewer 2 Report
The authors reported a simple technique to increase the photo efficiency for PID by simply deposition a thin layer of metals. All of my questions were addressed and the quality was improved. It can be published in Sensors.